# An Iron Refractory Phenotype in Obese Adipose Tissue Macrophages Leads to Adipocyte Iron Overload

**DOI:** 10.3390/ijms23137417

**Published:** 2022-07-03

**Authors:** Magdalene K. Ameka, William N. Beavers, Ciara M. Shaver, Lorraine B. Ware, Vern Eric Kerchberger, Kelly Q. Schoenfelt, Lili Sun, Tatsuki Koyama, Eric P. Skaar, Lev Becker, Alyssa H. Hasty

**Affiliations:** 1Department of Molecular Physiology and Biophysics, School of Medicine, Vanderbilt University, Nashville, TN 37212, USA; magdalene.ameka@vanderbilt.edu; 2Department of Pathobiological Sciences, School of Veterinary Medicine, Louisiana State University, Baton Rouge, LA 70803, USA; wbeavers@lsu.edu; 3Division of Allergy, Pulmonary, and Critical Care Medicine, Vanderbilt University Medical Center, Nashville, TN 37212, USA; ciara.shaver@vumc.org (C.M.S.); lorraine.ware@vumc.org (L.B.W.); vern.e.kerchberger@vumc.org (V.E.K.); 4Department of Pathology, Microbiology, and Immunology, School of Medicine, Vanderbilt University Medical Center, Nashville, TN 37212, USA; eric.skaar@vumc.org; 5Department of Cancer Research, University of Chicago, Chicago, IL 60637, USA; kelly1@uchicago.edu (K.Q.S.); levb@uchicago.edu (L.B.); 6Department of Biostatistics, Vanderbilt University Medical Center, Nashville, TN 37212, USA; lili.sun@vumc.org (L.S.); tatsuki.koyama@vumc.org (T.K.); 7VA Tennessee Valley Healthcare System, Nashville, TN 37212, USA

**Keywords:** immunometabolism, polarization, obesity, iron, adipose tissue macrophage

## Abstract

Adipocyte iron overload is a maladaptation associated with obesity and insulin resistance. The objective of the current study was to determine whether and how adipose tissue macrophages (ATMs) regulate adipocyte iron concentrations and whether this is impacted by obesity. Using bone marrow-derived macrophages (BMDMs) polarized to M0, M1, M2, or metabolically activated (MMe) phenotypes, we showed that MMe BMDMs and ATMs from obese mice have reduced expression of several iron-related proteins. Furthermore, the bioenergetic response to iron in obese ATMs was hampered. ATMs from iron-injected lean mice increased their glycolytic and respiratory capacities, thus maintaining metabolic flexibility, while ATMs from obese mice did not. Using an isotope-based system, we found that iron exchange between BMDMs and adipocytes was regulated by macrophage phenotype. At the end of the co-culture, MMe macrophages transferred and received more iron from adipocytes than M0, M1, and M2 macrophages. This culminated in a decrease in total iron in MMe macrophages and an increase in total iron in adipocytes compared with M2 macrophages. Taken together, in the MMe condition, the redistribution of iron is biased toward macrophage iron deficiency and simultaneous adipocyte iron overload. These data suggest that obesity changes the communication of iron between adipocytes and macrophages and that rectifying this iron communication channel may be a novel therapeutic target to alleviate insulin resistance.

## 1. Introduction

Iron overload is correlated with metabolic dysfunction in a condition known as dysmetabolic iron overload syndrome [1]. Adipose tissue iron overload is a feature of multiple models of obesity and insulin resistance (IR; reviewed in [2]). Iron overload causes IR in all metabolic tissues by a variety of mechanisms [3]. Adipocyte iron overload increases inflammation [4,5], decreases insulin sensitivity [6,7,8], alters adipokine release [9,10], and regulates energy homeostasis through leptin [10]. In general, excess iron leads to the generation of damaging reactive oxygen species and iron-induced apoptosis, called “ferroptosis” [11]. Excess iron in adipocytes may lead to lipid peroxidation, a process that generates free radical peroxyl species, which can impair insulin signaling [12]. The factors that contribute to adipocyte iron overload during obesity are unknown; however, the demand and turnover of iron is presumably low in mature adipocytes [13,14], suggesting that another cell type within adipose tissue could contribute to adipocyte iron overload.

Almost all known immune cells can be found in adipose tissue, and their proportions and inflammatory status change in obese compared to lean settings [15]. Of these, we focused on the role of macrophages in controlling adipocyte iron homeostasis, for the following reasons. First, macrophages regulate iron bioavailability systemically by either donating iron or buffering excess iron from cells as needed [16]. Second, at the tissue level, macrophages have been shown to control local iron concentrations [17]. For example, in wound healing, macrophages sequester iron during the inflammatory phase to prevent pathogen proliferation, but when the wound converts to the healing phase, the macrophages change their phenotype and provide iron needed for tissue repair [18]. Third, we previously discovered a specialized iron-handling macrophage in adipose tissue (termed MFe^hi^), with increased inflammatory gene expression and reduced iron-handling capacity in obese settings [19]. This dysfunctional MFe^hi^ iron handling is coincident with adipocyte iron overload [19]. Furthermore, MFe^hi^ can take up the excess iron, preventing adipocyte iron overload in lean conditions [20]. Finally, using different models of obesity [21,22], others have observed a decrease in the expression of discrete iron trafficking proteins in ATMs; however, these studies did not assess how obesity impacts the wide range of compensatory iron trafficking proteins and net iron communication between ATMs and adipocytes. 

Adipose tissue macrophages (ATMs) are uniquely polarized to adapt to a lipid-rich environment. This unique polarization is characterized by both inflammatory gene expression, albeit at levels lower than those of classical LPS-stimulated M1 macrophages, as well as upregulation of proteins involved in lipid turnover [6,21]. There have been a number of terms suggested for these ATMs [6,21,23,24]; we refer to them as metabolically activated or “MMe” [21], a polarization state that can be reproduced in culture with glucose, insulin, and palmitic acid. Whether the distinction between M1 and obesogenic MMe macrophage polarization states extends to their role in regulating adipose tissue iron homeostasis, and how this relates to adipocyte iron overload, is unknown. However, inferences can be made based on established patterns in M1-polarized macrophages. In response to pathogens, pro-inflammatory conditions drive M1 macrophages to sequester iron. This triggers an inflammatory program in the macrophages and reduces iron availability to extracellular microbes, which use iron for their own replication, survival, and virulence [16]. If the inflammatory MMe macrophages model this same behavior, they would sequester iron and would not contribute to adipocyte iron overload. However, if MMe macrophages have an iron-donating or iron-recycling phenotype similar to M2-like macrophages [25,26], they could donate iron to adipocytes and contribute to adipocyte iron overload in obesity.

Herein, we tested the hypothesis that MMe macrophages have a unique iron-handling profile, distinct from M1 and M2, and that they contribute to adipocyte iron overload in obesity. First, we evaluated the effects of the obese environment on the management of iron in ATMs. Next, we assessed the bioenergetic function of lean and obese ATMs in response to iron. Finally, we explored how the communication of iron signals from MMe macrophages to surrounding adipocytes is altered in obesity. Our data show an iron refractory phenotype within obese ATMs that directly contributes to adipocyte iron overload. 

## 2. Results

### 2.1. Obesogenic ATMs Have Reduced Expression of Iron-Trafficking Proteins 

Adipocyte iron overload accompanies obesity and IR, despite several counter-regulatory cues [9,10,27]. We hypothesized that ATMs contribute to this adipocyte iron overload. We first set out to understand iron handling of obese ATMs. Previous work showed that ATMs in obese adipose tissue have a unique MMe phenotype [21,28]. We modeled this state in vitro by treating BMDMs with palmitic acid, glucose, and insulin as previously reported [21]. M0, M1, and M2 polarization states were included in our analyses, with M1 modeling inflammatory macrophages and M2 modeling resident ATMs in lean adipose. 

Our cell treatments led to successful polarization, as shown by qRT-PCR analysis of genes associated with each polarization state (Figure 1A). We then assessed iron-handling genes (Figure 1B) and found M1 polarized cells had increased ferritin heavy chain (*Fth1*) and decreased ferroportin (*Slc40a1*) compared to M0 BMDMs, indicating an iron storage phenotype. In contrast, M2 macrophages had increased expression of iron storage (*Fth1* and *Ftl1*), processing (heme oxygenase (*Hmox1*)), and import (transferrin receptor (*Tfr1*), low density lipoprotein receptor (*Cd91*), and the hemoglobin haptoglobin receptor (*Cd163*)) genes. This expression pattern indicates a recycling phenotype, as has been reported [25,26]. Compared to M1 and M2 macrophages, MMe macrophages had lower expression of most iron-handling genes quantified. 

Protein analyses generally supported the gene expression data for the M1 and M2 BMDMs and aligned closely with reported observations [29]. MMe macrophages displayed a consistently lower expression of iron import, storage, and export proteins (Figure 1C,D) compared to the other macrophage types. Using LC-MS/MS, Becker et al. observed a similar negative trend in the expression of some iron-related proteins with MMe polarization in mice and human macrophages [21,28] and reproduced this phenotype using whole cell lysates from polarized mouse macrophages (Appendix A). TFRC and FTL1 were both lower in MMe macrophages than in other polarization states. To further support a difference in iron handling between M1 and MMe macrophages, MMe macrophages retained less iron than M1 macrophages (Figure 1E).

In our in vivo model, mice were fed a low-fat diet (LFD) or a high-fat diet (HFD) for 16 weeks, and ATMs were isolated. The trend of blunted expression of iron-handling proteins seen in MMe macrophages was recapitulated in F4/80^+^/CD11b^+^ ATMs isolated from obese mice (Appendix A). In the analyses of microarray data, Ferrante et al. also observed similar trends: a diminution in transcription of the genes involved in iron import and export in sorted ATMs from obese leptin deficient mouse models (GEO Accession number GSE53403) [22]. Therefore, using multiple approaches, we observe a reduction of iron-related genes and proteins in macrophages with metabolic activation. 

### 2.2. ATMs from Obese Mice Are Impervious to Excess Iron 

To model a biologically relevant source of iron that is modulated in obese IR conditions, we injected mice with a sub-lethal dose of cell-free hemoglobin (Hgb). Fluorescent Hgb was readily taken up by F4/80^+^ ATMs in lean chow-fed mice (Figure 2A). Interestingly, while ATMs from LFD-fed mice readily took up additional iron, ATMs from HFD-fed mice did not (Figure 2B). Of clinical relevance, septic patients with a BMI > 25 had higher concentrations of plasma-free Hgb compared to lean patients (Figure 2C).

### 2.3. Loss of Metabolic Flexibility in Iron-Challenged Obese ATMs 

Given that proper mitochondrial function is critical for macrophage plasticity [30], and that iron plays a key role in regulating mitochondrial and immune function [31,32], we assessed cellular respiration as a gauge of macrophage plasticity. Patterns of extracellular acidification rate (ECAR) and oxygen consumption (OCR) are shown in Figure 3A and Figure 4A. In response to an influx of a non-toxic dose of labile iron, cells with adaptive capacity increased their metabolic rates to restore redox balance [33]. ATMs from lean mice remained flexible and increased their basal (Figure 3B) and glucose-stimulated glycolytic rates (Figure 3C) as well as glycolytic capacities (Figure 3D) in the presence of labile iron.

Although ATMs from obese mice started out at a higher ECAR, as previously reported [34], they lost their ability to adapt to excess iron. Therefore, basal glycolytic rates and glucose-stimulated glycolytic rates did not increase in response to iron (Figure 3B,C) and glycolytic capacity was reduced (Figure 3D) in ATMs from obese mice injected with iron. 

Similar to work by Serbulea et al. [34], maximal respiratory capacity and spare capacity were increased by obese conditions. ATMs from lean mice were remarkably sensitive to iron treatments and greatly increased oxygen consumption rates in response to Hgb compared to obese mice (Figure 4A). ATP-linked respiration and non-mitochondrial respiration increased equally in iron-challenged ATMs from LFD and HFD-fed mice (Figure 4B,C), while maximal respiratory capacity (Figure 4D) and spare capacity (Figure 4E) fell in obese iron-challenged ATMs compared to lean iron treated ATMs and did not change compared to saline-treated ATMs from obese mice. 

### 2.4. Iron Exchange Is Enhanced between MMe Macrophages and Adipocytes, Culminating in Adipocyte Iron Overload

To understand how iron could be trafficked between macrophages and adipocytes during obesity, we developed an iron isotope labelled co-culture system that exploits mass differences in low-abundance iron isotopes to track and quantify iron (model diagram in Figure 5). Primary cells were differentiated and polarized with continuous enrichment with different low-abundance iron isotopes. BMDMs were labeled with ^54^Fe and adipocytes with ^57^Fe, and ^58^Fe was added to the media at the beginning of the 48-h co-culture period. This system allowed for the unbiased accounting of the origin of iron isotopes in the cells and media after co-culture. Before co-culture, differentiated macrophages and adipocytes were more than 90% loaded with their respective iron isotope (Appendix A). The cells were kept in separate chambers of a transwell in the ^58^Fe-labeled media for the 48-h co-culture period. At the end of the co-culture, macrophages, adipocytes, and media were collected and analyzed separately for all three isotopes. 

After co-culture with adipocytes, M0, M1, and M2 macrophages had similar amounts of ^54^Fe remaining, while MMe macrophages had a significant reduction of intracellular ^54^Fe (Figure 6A), despite loading with equal concentrations of ^54^Fe at the beginning of the co-culture. In contrast, the MMe macrophages had a significant 9-fold increase in ^57^Fe taken up from the adipocytes (Figure 6B). Adipocytes co-cultured with M2 and MMe macrophages had the least amount of their original ^57^Fe iron remaining (Figure 6C). Furthermore, MMe macrophages transferred approximately five times more iron to co-cultured adipocytes (Figure 6D) compared to the other polarization states. 

Almost no ^58^Fe from the media was taken up by the macrophages or the adipocytes in the co-culture system (data not shown). ^58^Fe remaining in the media was significantly different based upon the polarization state of the macrophages (Figure 6E), although these differences were quantitatively much lower than those in the macrophages and adipocytes. Concerning efflux of iron from macrophages and adipocytes to the media, adipocytes contributed some amount of ^57^Fe to the surrounding media, and the highest contribution came from adipocytes co-cultured with MMe macrophages as opposed to those co-cultured with M1 and M2 macrophages (Figure 6F). There was no ^54^Fe (from macrophages) detected in the media (data not shown).

Using ICP-MS, it was also possible to quantify ^56^Fe remaining in cells and media. Interestingly, ^56^Fe was only detected in media. By summing the total iron in the macrophages, adipocytes, and media, an interesting relationship between macrophages and adipocytes in obesity was uncovered (Figure 6G and Appendix A). While the MMe macrophages have the least amount of iron at the end of the co-culture, they receive the largest proportion of iron from surrounding adipocytes. Additionally, adipocytes co-cultured with MMe macrophages had a significant increase in iron compared to those co-cultured with the M2 polarized macrophages. 

## 3. Discussion

### 3.1. Summary

Our data show, for the first time, that iron can be transferred between macrophages and adipocytes. Furthermore, we show that the net accumulation of iron in the adipocytes is greater when they are co-cultured with metabolically activated macrophages that model ATMs recruited to obese adipose tissue as compared to M2-like-macrophages that model resident ATMs in lean adipose tissue. 

### 3.2. Iron Homeostasis in Obese ATMs

Fine-tuned control of tissue iron concentrations is pivotal for optimal function, with iron overload being considered particularly detrimental due to ferrotoxic damage [35]. Macrophages, which are the main iron cycling cells [13,14], are already resident and ubiquitous in adipose tissue and rapidly increase in number with obesity [36,37]. Therefore, it is plausible that macrophages in adipose tissue play a role in modulating adipose tissue iron concentrations in lean and obese conditions. Our own published data support this [19,20]. We discovered a specialized population of alternatively activated tissue resident ATMs with a high iron handling capacity (MFe^hi^) from lean mice that could be sorted due to magnetic pull. We found that, during obesity, as the adipocytes became iron loaded, these MFe^hi^ ATMs became more inflammatory, lost expression of a number of iron-related genes, and were replaced by low-capacity MFe^lo^ ATMs. Additionally, all ATMs recruited during obesity were MFe^lo^. The actions of infiltrating metabolically activated ATMs mediate much of the maladaptation associated with obesity. Therefore, we focused our current work on understanding how iron is regulated in this macrophage type with the eventual goal of understanding whether dysregulation at this node would explain how adipocytes becomes iron overloaded in obesity.

Our current studies show that, much like MFe^hi^ ATMs, MMe macrophages and obese ATMs lose expression of iron-related genes and proteins. However, unlike MFe^hi^ ATMs, the obese ATMs are inherently inflammatory and have the lowest amount of iron, even when challenged with exogenous iron. In fact, they appear to be more like the MFe^lo^ cells from obese mice, which trended toward lower iron content. We contend that, in this sense, the MFe^lo^ ATMs recruited during obesity may be the same cell type as the MMe obese ATMs, but further experimentation would be required to validate this. 

It should be noted that the mRNA and protein data are not exactly concordant for all iron-related genes/proteins we quantified. This is not unusual, as iron protein concentrations are also strongly controlled at the post-transcriptional level by the iron response element (IRE)/iron response protein (IRP) system [38]. The effects of obesity on the iron-sensing IRE/IRP system of ATMs were not assessed in this study but may give pivotal insights into the dysregulation of iron-related proteins in obese ATMs. The consideration of the modulation of the IRE/IRP system with obesity is especially enticing given the iron-dependent dual role of IRP1 as a cytosolic aconitase. Assessing how the IRE/IRP system modulates expression of different iron- and metabolic-related proteins with obesity would outline targets that bridge cytosolic respiration to fine-tuned iron sensing in obese ATMs. None-the-less, our data demonstrate consistently lower iron-handling genes, proteins, and iron concentrations in MMe macrophages and in ATMs from obese mice. 

### 3.3. Modes of Cellular Respiration in Iron-Challenged Lean and Obese ATMs

The results of our modified mitochondrial stress test (MMST) demonstrated the profound sensitivity of ATMs to iron treatments. This is not surprising given the lipid-rich niche of ATMs that demands fine-tuned control of iron concentrations to avoid iron-related oxidative stress. Lean ATMs are thought to be alternatively activated [39], and this phenotype matched our expectations in that this macrophage type was highly responsive to changes in local iron concentrations [25,26]. Therefore, lean ATMs mitigate damage from excessive iron concentrations, either because of or by way of, increasing their bioenergetic bandwidth. Our MMST emphasize the unique nature of obese ATMs. These ATMs are already filled with lipids and are respiring at a high rate through both glycolysis and OXPHOS [34]. The expansion of bioenergetic bandwidth with iron treatment is largely lost in obese ATMs, and only ATP-linked and non-mitochondrial measures of respiration are increased with iron treatment in obese ATMs. Interestingly, in obese ATMs, respiratory states that are upregulated by lipids are downregulated by iron treatments, and those not affected by lipids are sensitive to iron regulation. Unlike lean ATMs, glycolytic capacity is reduced. This points to a unique energetic crisis that must be occurring in the mitochondria of obese iron-challenged ATMs. In the presence of iron, a rich source of ready electrons that can power the electron transport chain, obese ATMs do not rely exclusively on increasing mitochondrial respiration and respiratory capacity, which would be the safest and most efficacious way to deal with increased oxidative stress. Instead, low-capacity obese ATMs bifurcate their mode of respiration and use both ATP synthase and non-mitochondrial means for energy production. The results suggest that increased use of glycolytic pathways is not involved to buffer iron excess. The consequences of the shift in bioenergetics away from glycolysis to favor another non-mitochondrial process warrants further studies. 

### 3.4. Iron Exchange in Macrophage and Adipocyte Co-Cultures 

Our previous work has suggested, but not proven, that iron can be exchanged between ATMs and adipocytes, and that this exchange is influenced by the macrophage phenotype [19,20]. In the current study, we tested this hypothesis using isotope-labeled macrophage and adipocyte co-cultures. These data are, to our knowledge, the first to directly show iron transfer between macrophages and adipocytes. Furthermore, we clearly show that MMe macrophages can release more of their iron pool and can take up nine times as much iron from adipocytes in the co-culture compared to all other polarization states (Figure 6B). Adipocytes also take up five times more iron when co-cultured with MMe macrophages compared to the other polarization states (Figure 6D). These data suggest that reciprocal iron transfer between macrophages and adipocytes is enhanced in obese conditions. Importantly, the final outcome of this exchange is a two-fold increase in total iron in adipocytes cultured with MMe macrophages compared to those cultured with M2 macrophages (Figure 6G). This suggests that MMe macrophages in obese adipose tissue contribute to adipocyte iron overload in obesity. 

While the lack of ^58^Fe, coming from the media, in macrophages and adipocytes at the end of the co-culture was unexpected, it was not due to technical difficulties. In additional control experiments, we confirm that this was not due to an inability of macrophages to take up ^58^Fe. ^54^Fe-loaded macrophages took up ^58^Fe when the cells were cultured on their own (i.e., not co-cultured) in ^58^Fe-labeled media for 48 h, reaching 15–25% of their total iron content (Appendix A). ^57^Fe loaded adipocytes, however, it did not take up ^58^Fe from the media after loading with ^57^Fe during their differentiation, suggesting that, after full differentiation, they do not readily take up iron from media. A more interesting interpretation of the observation that the differentiated cells in co-culture do not take up iron from the media is that there is an active selection bias in the cells in the co-culture to acquire iron from each other rather than from other iron-rich sources such as the media. For example, iron can be transferred in exosomes [40], acidic lysosomes [41], or extruded mitochondria [42], precluding the need for dedicated protein chaperones. Although there is no evidence that adipocytes produce siderophores, this is another intriguing potential mechanism for iron transfer between these two cell types. Future studies are needed to distinguish between these possibilities. 

Cumulatively, our data show that obese ATMs respond and adapt inappropriately to iron and lose their iron-buffering capacity. Additionally, there is an increased permissiveness to the exchange of iron between ATMs and adipocytes in obesogenic conditions, and adipocytes become iron overloaded. Our results suggest that restoring iron responsiveness in the lipid-filled obese ATM, possibly at the expense of lipid handling, or finding the optimal division of labor between iron buffering and lipid handling in the obese ATM may be beneficial to alleviating iron overload and its downstream effects. 

## 4. Materials and Methods

***Reagents and antibodies***: Primer information, antibody concentrations, and sources are described in Appendix A. Direct-zol RNA prep kits were purchased from Zymo Research (Irvine, CA, USA), and the iScript cDNA synthesis kit was purchased from Bio-Rad (Hercules, CA, USA). Reagents and materials for liquid chromatography mass spectrometry (LC-MS) experiments are detailed in Cui et al. [43]. Cytokines IL-4, IL-13 and IFN-γ were purchased from R&D Systems (Minneapolis, MN, USA) and reconstituted according to the manufacturer’s instructions. Cell-free hemoglobin (CFH) was purchased from Cell Sciences (Newburyport, MA, USA). AF488 was purchased from Molecular Probes (San Jose, CA, USA) and used to label CFH. CFH (10 mg/mL) was labeled with AF488 according to the manufacturer’s protocol. Excess dye was then removed by passing the solution through a Sephadex 25 column, which was eluted. Labeled CHF was stored in the dark at 4⁰C until use. SDS-PAGE gels and PVDF were purchased from Bio-Rad. All plates, mitochondrial inhibitors, and reagents for extracellular flux analysis were purchased from Agilent Technologies (Santa Clara, CA, USA). Insulin (Novolin R) for cell treatments was purchased from Novo Nordisk (Denmark). Lipopolysaccharide (LPS), palmitic acid (PA), DMEM, Penicillin/Streptomycin, HEPES, type II collagenase, RIPA lysis buffer, 100X HALT protease inhibitor cocktail, Trizol, haptoglobin, Trace Metal Grade 6 N HCl, and 70% OPTIMA Grade HNO_3_ were purchased from Thermo Fisher Scientific Brands (Waltham, MA USA). ^54^Fe_2_O_3_, ^57^Fe_2_O_3_, and ^58^Fe_2_O_3_ isotopes were purchased from Cambridge Isotopes (Tewksbury, MA, USA). 

***Mice and diets***: All animal care procedures received approval from and followed the guidelines of the Vanderbilt University Institutional Animal Care and Use Committee. Experiments used male 7–30-week-old C57BL/6J mice from The Jackson Laboratory (Bar Harbor, ME, USA). Mice were fed on chow, low-fat (LFD, D12450B) and high-fat (HFD, D12492) diets for 16 or 20 weeks (mouse diets from RD diets). For mice treated with hemoglobin, mice were either injected (IP) with saline (Sal) or with a sub-septic dose of hemoglobin (Hgb, 0.1 mg/kg cell free hemoglobin) every other day for 1 week for a total of four doses on the last week of dietary manipulation. 

***Cell culture and cell treatments***: Primary bone marrow-derived macrophage (BMDMs) from hematopoietic stem cells were isolated from the femurs and tibia of C57BL/6J mice and differentiated as described [44] with the following modifications. ACK-lysed bone marrow was quenched with 15% L929 supplemented media (L929 SM, L929-cell conditioned media is harvested and used to supplement DMEM for differentiation of BMDMs). Quenched cells were washed and seeded into 75 cc flasks. On day 6, fully differentiated BMDMs were split, either into 25 cc tissue culture flasks, Seahorse plates, or transwell inserts and maintained in 15% L929 SM. Cells were allowed to adhere for at least 24 h and were then polarized as follows: For unpolarized M0: no cytokines, M1 polarization: 24 h treatment with IFN-γ (0.1 μg/mL) and LPS (0.1 μg/mL), M2 polarization: 96 h treatment with IL-4 (0.1 μg/mL) and IL-13 (0.1 μg/mL), and for MMe polarization: 24 h treatment with 30 mM glucose, 0.08 pg/mL insulin (Novolin R), and Palmitic Acid (PA: 100 μM). All primary BMDMs were maintained in L929 SM: DMEM containing 10% (*v*/*v*) charcoal-stripped fetal bovine serum, 1% penicillin/streptomycin, 1% HEPES, and 15% L929-cell conditioned media at 37 °C in 5% CO2. Primary murine adipocytes were cultured as previously described [45] and maintained in adipocyte maintenance media until the day of co-culture. 

***Human cell-free hemoglobin measurements***: Plasma samples were collected from adult (≥18 year old) patients admitted to the intensive care unit at Vanderbilt University Medical Center with sepsis (Sepsis II criteria) [46] as part of the prospective observational VALID (Validating Acute Lung Injury marks for Diagnosis) study [47]. The study protocol was approved by the Vanderbilt University Institutional Review Board. We have previously reported plasma CFH levels measured by HemoCue for these patients [48]. 

***Isolation, magnetic sorting and fluorescence-activated cell sorting of adipose tissue macrophages*:** Methods were adapted and modified from [20]. ATMs were isolated exclusively from the epididymal adipose tissue (eAT) of male mice. The fat pads were dissected, minced, and digested in 3 mL of 0.2 mg/mL type II collagenase solution in a 0.5% BSA/DPBS for 20 min at 37 °C. The stromal vascular fraction (SVF) was separated from adipocytes by centrifugation at 700× *g* for 7 min, followed by erythrocyte lysis with ACK buffer and quenching in L929 SM. Lysed cells were filtered, and ATMs were sorted via FACS (for Western blotting) or magnetic bead sorting (for extracellular flux analysis). For FACS, the ACK-lysed SVF was treated with Fc block for 10 min and then stained with fluorophore-conjugated antibodies against cell-surface markers for 30 min at 4 °C at 1:200. Cells were washed in FACS buffer and resuspended and filtered to make a single-cell suspension and stained with DAPI. Live F4/80+/CD11b+ ATMs were sorted with a FACSAria III cell sorter (BD Biosciences) with appropriate compensation and fluorescence minus one controls. For magnetic sorting, ACK-lysed SVF was incubated with a 1:10 mix of F4/80+ magnetic beads (Miltenyi) for 15 min at 4 °C. Either the positive selection (possel) program on the AutoMACS magnetic activated cells-sorting system (Miltenyi, Auburn, CA, USA) or magnetic fields from the MACS MultiStand benchtop magnet (Miltenyi) were employed to separate F4/80+ ATMs according to manufacturer’s instructions. After separation and elution, the magnetic cells were separated by centrifugation at 700× *g* for 7 min at 4 °C and resuspended in nitric acid for digestion and ICP-MS. 

***qRT-PCR analysis***: RNA was isolated from polarized BMDMs and adipocytes using Trizol and RNA prep kits according to the manufacturer’s instructions. cDNA was synthesized according to the manufacturer’s instructions, and real-time (RT-qPCR) was performed on an iQ5 cycler (BioRad) using Taqman gene expression assays (Applied Biosystems via ThermoFisher). Primer sequences are available in Appendix A. Gene expression was analyzed using the 2-ΔΔCt method and normalized to the averages of beta-2-Microglobulin (B2M), with either β-actin (ACTB) or glyceraldehyde-3-phosphate dehydrogenase (GAPDH) according to [49].

***Western blot analysis***: Polarized BMDMs, isolated ATMs, and adipocytes were lysed in RIPA lysis buffer containing HALT, 1 mM sodium orthovanadate, and 1 mM PMSF. Cells were lysed by centrifugation at 14,800× *g* for 12 min at 4 °C. Protein was quantified from whole cell lysates using a BCA assay (Pierce, Dallas, TX, USA). Subsequently, 50–150 ng of protein was electrophoresed through 4–20% SDS-PAGE gels, transferred onto a low-fluorescence PVDF membrane, and immunoblotted as described. Antibody information is available in Appendix A. Blots were developed using SuperSignal™ West Femto Maximum Sensitivity Substrate and visualized using Gel Doc EZ Gel Documentation System (Bio-Rad). Band intensity densitometry was quantified using Image J (NIH, Bethesda, MD, USA) and normalized to β-actin.

***Inductively-coupled Mass Spectrometry (ICP-MS)***: For inductively-coupled mass spectrometry (ICP-MS), elemental quantification on acid-digested samples (diluted to 2% HNO_3_) was performed using an Agilent 7700 inductively-coupled plasma mass spectrometer (Agilent, Santa Clara, CA, USA) attached to a Teledyne (CETAC Technologies ASX-560) autosampler (Teledyne CETAC Technologies, Omaha, NE, USA). The following settings were fixed for the analysis: Cell Entrance = −40 V, Cell Exit = −60 V, Plate Bias = −60 V, OctP Bias = −18 V, and collision cell Helium Flow = 4.5 mL/min. Optimal voltages for Extract 2, Omega Bias, Omega Lens, OctP RF, and Deflect were determined empirically before each sample set was analyzed. ^54^Fe_2_O_3_, ^57^Fe_2_O_3_, and ^58^Fe_2_O_3_ (Cambridge Isotope Laboratories, Tewksbury, MA, USA) were dissolved to 100 mM in Trace Metal Grade 6 N HCl. A calibration curve for each isotope was made in 2% OPTIMA Grade HNO_3_ in Ultrapure water at 0, 1, 10, 100, 1000, and 10,000 ppb. Samples were introduced by peristaltic pump with 0.5 mm internal diameter tubing through a MicroMist borosilicate glass nebulizer (Agilent). Samples were initially taken up at 0.5 rps for 30 s followed by 30 s at 0.1 rps to stabilize the signal. Samples were analyzed in Spectrum mode at 0.1 rps, collecting three points across each peak and performing three replicates of 100 sweeps for each element analyzed. The sampling probe and tubing were rinsed for 20 s at 0.5 rps with 2% nitric acid between every sample. Data were acquired and analyzed using the Agilent Mass Hunter Workstation Software version A.01.02. 

***Liquid Chromatography-Mass Spectrometry***: LC-MS in polarized human and mouse macrophages was performed as previously described [43].

***Extracellular flux analysis***: A modified mitochondrial stress test (MMST) was developed in consultation with Agilent (Santa Clara, CA, USA) specialists and was performed on ATMs. Extracellular flux analysis to monitor mitochondrial oxygen consumption rate (via OCR) and glycolytic rates (from extracellular acidification rate (ECAR)) were measured via a Seahorse XFe96 Extracellular Flux Analyzer. Our modification combined glycolytic and respiratory measurements in one assay measuring ECAR concurrently with OCR in one test. The first section of the test measures glycolysis and uses unbuffered reagents as recommended for a standard glycolytic stress test. Post-oligomycin, the rest of the test measures mitochondrial respiration only by using specific mitochondrial targeted inhibitors. As a result, only the appropriately measured respiratory states were calculated from each section of the MMST, and non-mitochondrial oxygen consumption can only be approximated because basal OCR was not measured in our test. Additionally, we are unable to measure glycolytic reserve as post 2-DG ECAR readings were not included in the test. ATMs from Sal and Hgb-treated animals on the same diet were seeded onto the same XFe96-well Seahorse plate. The entire outer rim was filled with media only, without cells for background correction. After 24 h to allow cells to adhere, ATMs were gently washed with XF Seahorse media (Agilent, pH 7.4) at least three times. Media was replaced with XF Seahorse media supplemented with 2 mM L-glutamine at pH 7.4 and allowed to equilibrate for 2–4 h at 37 °C without CO_2_ before starting the stress test. The following specifications were added to the MMST. Three measurements were recorded in 6–7 min intervals in each reagent. The first three points of flux were derived from basal glutamine utilization. An amount of 10mM of glucose was injected as a substrate for glycolytic analysis, and then mitochondrial inhibitory drugs for a mitochondrial stress test (MST) were added as follows: 1.5 µM oligomycin, 1.5 µM FCCP, 0.75 µM rotenone/antimycin. ECARs and OCRs at each time point were averaged for each reagent and for each animal. Average values for different parameters of mitochondrial function were measured from parallel experiments in which the ATMs came from mice on the same diet that received identical treatments. ECAR and OCR rates were normalized to cellular protein content calculated by BCA. 

Specialized glycolytic ECAR rates were calculated as follows:Basal ECAR = Glutamine-linked ECAR rateGlycolytic rate = ECAR rates after addition of glucoseGlycolytic capacity estimates = ECAR rates after addition of oligomycin.

To evaluate coupled respiration, respiratory states were calculated as follows: Non-mitochondrial oxygen consumption rate estimate = OCR rates after Rot/AA inhibitionATP-linked respiration = oligomycin inhibited OCR ratesMaximal respiration rate = FCCP-inhibited OCR rates 

Spare capacity = FCCP-inhibited OCR rates–oligomycin inhibited OCR rates

***Iron isotope labeling in the macrophage/adipocyte iron transfer system***: Primary adipocytes and primary BMDMs were cultured as described in [45] and above. Steps were followed according to the schematic shown in Figure 5. Primary adipocytes were maintained in ^57^Fe. BMDMs were matured and polarized separately and maintained in ^54^Fe. Isotopic iron (5 µM) was used in all cases. Isotopic iron was added to the respective culture media during differentiation, and cells were maintained with the isotope label (respective iron isotope added to all media changes) until the day of co-culture. Twenty-four hours before co-culture, polarizing cytokines were removed from BMDMs and were allowed to rest in complete L929 SM. On the day of co-culture, iron-loaded cells were washed twice with large volumes of sterile PBS and co-cultured in a transwell system in a 1:1 mix of the respective cell culture media spiked with ^58^Fe for 48 h as shown. At the end of the 48 h, cells were either harvested for ICP-MS or used in protein quantification by BCA. 

***Statistics***: For comparisons of gene and protein expression, natural and isotopic iron content in M0, M1, M2, and MMe polarized BMDMs and ^56^Fe levels in ATMs isolated from lean and obese mice, Student’s *t*-test or one-way analysis of variance (ANOVA) were used. Type I error rate control for multiplicity was achieved with the Bonferroni method when pair-wise comparisons were specified in advance, or Tukey’s method when all pair-wise comparisons were considered. For respiratory states in ATMs from Hgb-injected mice on low-fat and high-fat diets, two-way ANOVA with Tukey’s multiple comparison post-hoc tests were used. Comparison of CFH in obese (BMI > 25) to non-obese (BMI ≤ 25) patients was performed by Wilcoxon rank sum test, after excluding those with CFH levels that were <10 mg/dL (the lower limit of detection for HemoCue). The relationship between BMI and CFH was performed using linear regression analysis on 425 patient samples using R (version 4.1, Vienna, Austria).

## Figures and Tables

**Figure 1 ijms-23-07417-f001:**
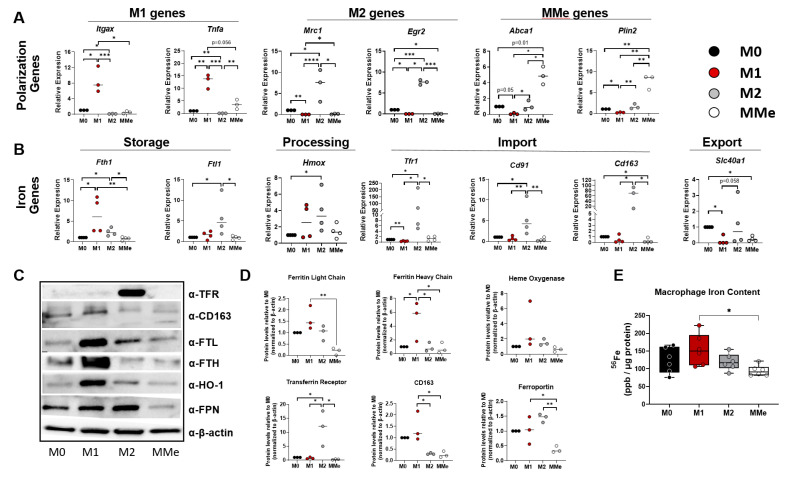
**Loss of iron-related gene and protein expression in obesogenic macrophages**. Primary bone marrow-derived macrophages (BMDMs) were differentiated and polarized into unpolarized (M0), M1, M2, and MMe states. (**A**) Expression of polarization genes. (**B**) Expression of iron-handling genes. (**C**,**D**) Western blot for iron-handling proteins and their quantification. (**E**) Iron (^56^Fe) content as measured by ICP-MS in polarized BMDMs. Statistics were performed using one-way ANOVA with Bonferroni or Tukey’s *post-hoc* tests for multiple comparisons. * *p* < 0.05; ** *p* < 0.01, *** *p* < 0.01, **** *p* < 0.001.

**Figure 2 ijms-23-07417-f002:**
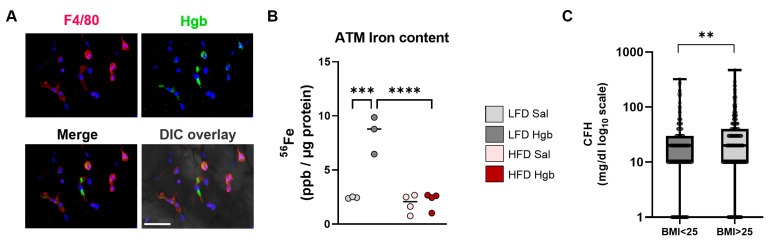
**Hemoglobin is taken up by ATMs, and cell-free hemoglobin concentrations change with obesity**. (**A**) Seven- to nine-week-old chow-fed male mice were injected IP with fluorescent hemoglobin, and adipose tissue slices from these mice were stained with F4/80 and DAPI (blue) and imaged. (**B**) Iron (^56^Fe) content from adipose tissue macrophages (ATMs) isolated from mice fed a low-fat diet (LFD) or a high-fat diet (HFD) for 20 weeks and injected with saline (Sal) or hemoglobin (Hgb) at week 19 every other day for a total of 4 doses. (**C**) Plasma samples were collected from human adults of both sexes with varying BMIs and cell-free hemoglobin (CFH) levels measured by HemoCue. For panel B, two-way ANOVA with Bonferroni’s multiple comparison *post-hoc* tests were used (*n* = 3). For Panel C, Wilcoxon rank sum test for CFH in humans (*n* = 425) was used. ** *p* < 0.01, *** *p* < 0.001, **** *p* < 0.0001.

**Figure 3 ijms-23-07417-f003:**
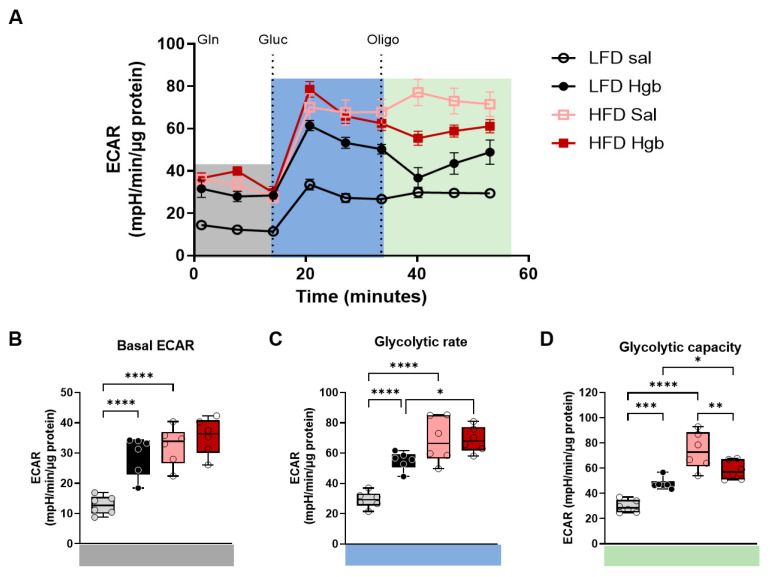
**Bioenergetic shifts in response to iron in obese adipose tissue macrophages**. Mice were fed low-fat (LFD) or high-fat (HFD) diets for 20 weeks. At week 19–20 of dietary challenge, mice were injected with either saline (Sal) or hemoglobin (Hgb). Adipose tissue macrophages (ATMs) were isolated using F4/80+ magnetic beads. (**A**) Extracellular acidification rate (ECAR) was measured by extracellular flux analysis in a modified mitochondrial stress test. Mitochondrial respiratory rates were calculated from extracellular flux analysis rates, as described in the Methods section. (**B**) Basal ECAR. (**C**) Glycolytic rate. (**D**) Glycolytic capcity. Statistics were performed using two-way ANOVA with Tukey’s multiple comparison *post-hoc* tests. * *p* < 0.05, ** *p* < 0.01, *** *p* < 0.001, **** *p* < 0.0001.

**Figure 4 ijms-23-07417-f004:**
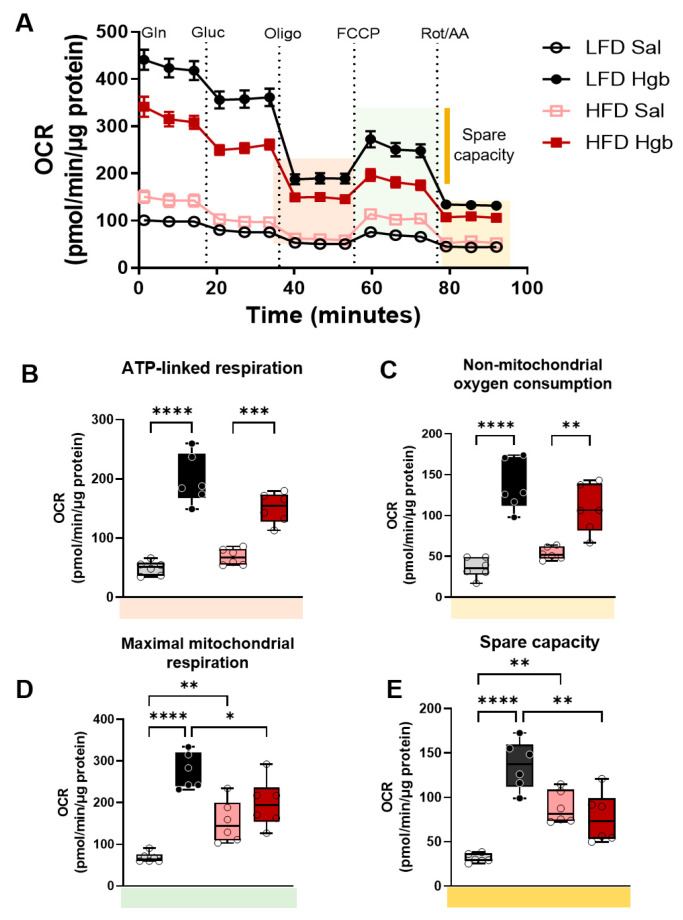
**Bioenergetic shifts in response to iron in obese adipose tissue macrophages**. Mice were fed a low-fat (LFD) or high-fat (HFD) diet for 20 weeks. At week 19–20 of dietary challenge, mice were injected with either saline (Sal) or hemoglobin (Hgb). Adipose tissue macrophages (ATMs) were isolated via F4/80+ magnetic beads. Mitochondrial respiratory rates were calculated from extracellular flux analysis rates, as described in the Methods section. (**A**) Oxygen consumption rate (OCR) was measured using a modified mitochondrial stress test. (**B**) ATP-linked respiration. (**C**) Non-mitochondrial oxygen consumption. (**D**) Maximal mitochondrial respiration. € Spare capacity. Statistics were performed using two-way ANOVA with Tukey’s multiple comparison *post-hoc* tests. * *p* < 0.05, ** *p* < 0.01, *** *p* < 0.001, **** *p* < 0.0001.

**Figure 5 ijms-23-07417-f005:**
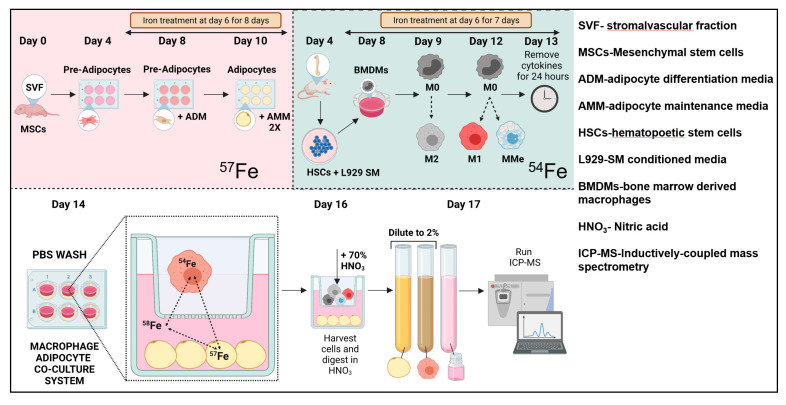
**Schematic describing the procedure and workflow for isotopic iron labeling co-culture system**.

**Figure 6 ijms-23-07417-f006:**
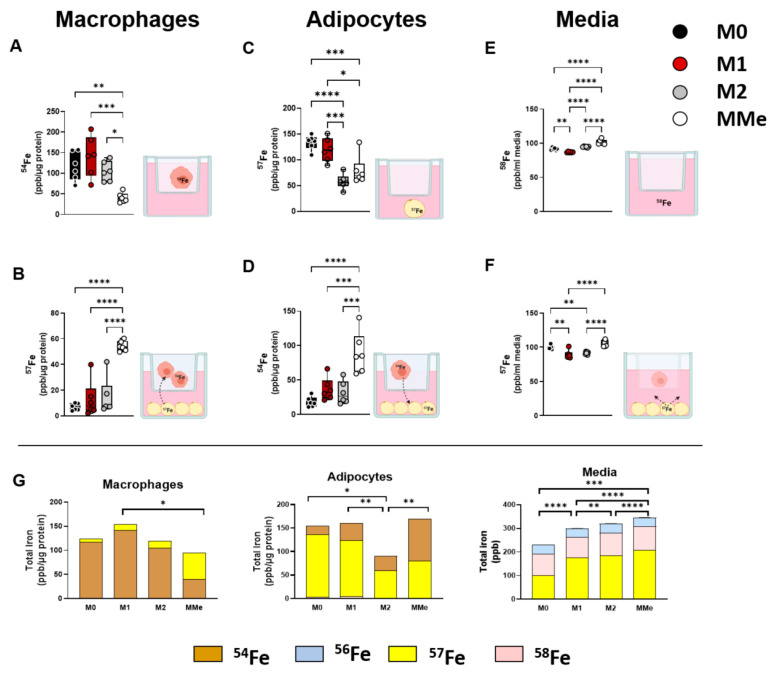
**Increased iron communication between macrophages and adipocytes in obesogenic conditions**. Macrophages were differentiated and polarized in ^54^Fe while adipocytes were differentiated in ^57^Fe. After complete differentiation and uptake of isotopes, macrophages and adipocytes were co-cultured for 48 h in media containing ^58^Fe. ^54^Fe, ^57^Fe, and ^58^Fe were measured in macrophages, adipocytes, and media at the end of the co-culture period. (**A**) ^54^Fe remaining in macrophages. (**B**) ^57^Fe transferred from adipocytes to macrophages. (**C**) ^57^Fe remaining in adipocytes, (**D**) ^54^Fe transferred from macrophages to adipocytes. (**E**) ^58^Fe remaining in media. (**F**) ^57^Fe transferred from adipocytes to media. (**G**) The total iron (^54^Fe, ^56^Fe, ^57^Fe, and ^58^Fe) in each cell type. Statistical analyses for all data were performed using two-way ANOVA with Tukey’s multiple comparison *post-hoc* tests for total iron only (individual isotopes are quantified in panels (**A**–**F**). * *p* < 0.05, ** *p* < 0.01, *** *p* < 0.001, **** *p* < 0.0001.

## Data Availability

Not applicable.

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
