# Peer review of "An Iron Refractory Phenotype in Obese Adipose Tissue Macrophages Leads to Adipocyte Iron Overload"

_ijms, 2022, doi:10.3390/ijms23137417_

Round 1

Reviewer 1 Report

Manuscript # 1787777

In the current manuscript titled “ An iron refractory phenotype in obese adipose tissue macro-2 phages leads to adipocyte iron overload” authors have tried to establish the interaction of adipose-derived macrophages (ATM) with adipocytes and the exchange of iron and maintain homeostasis. However, during obese conditions ATP iron handling mechanism in ATMs especially metabolically activated macrophages phenotype (MMe) gets disturbed which led to adipocyte's iron overload and associated Obesity and Insulin Resistance. This is interesting work to understand the role of macrophages (immune cells) in the development of iron overload in adipocytes and the development of metabolic diseases including obesity. However, I have the following comments for this study. 

Comments: 

1)    Although authors have tried to establish the role of adipose-derived macrophages (especially metabolically activated macrophages) in the development of adipocytes’  iron overload due to its iron handling machinery and related proteins. However, I was curious to know authors tried to establish the role of ATMs but no other immune cells and non-immune cells in adipocyte iron overload and disease progression?

2)    In figure 1 representation is not good for qPCR data. The font size is very small which makes reader too difficult to read the labeling. I would suggest increasing the Font size.

3)    In Figure, WB data and qPCR data is not matching. The Authors have shown that Iron regulating genes including ftl1, and cd163 were increased in M2 macrophages in qPCR data, however, expression of these genes increased in the M1 microphase shown in WB. This is contracting data. Along with this, there is no significant change in the heme oxygenase gene in M1 macrophages, however, there is a significantly increased expression of HO protein in M1 macrophages. In addition to this, there is no quantification for WB.

4)    Authors are talking about Figures in the text but there is no figure 6. This is really a big mistake. It seems that proofreading was not done by the authors. Authors should work on this.

5)    Font size of the text is different from lines 169 -181 and 300-316. The authors should correct this.

6)   Heading of material methods and Result section is not bold. It is not distinct and mixes with regular text. (For example Cell culture and cell treatments: Primary bone marrow-derived macrophage and other sections).

7)    Typo error in text line 147 (plasma Vfree Hgb compared…)

Author Response

In the current manuscript titled “An iron refractory phenotype in obese adipose tissue macrophages leads to adipocyte iron overload” authors have tried to establish the interaction of adipose-derived macrophages (ATM) with adipocytes and the exchange of iron and maintain homeostasis. However, during obese conditions ATP iron handling mechanism in ATMs especially metabolically activated macrophages phenotype (MMe) gets disturbed which led to adipocyte's iron overload and associated Obesity and Insulin Resistance. This is interesting work to understand the role of macrophages (immune cells) in the development of iron overload in adipocytes and the development of metabolic diseases including obesity. However, I have the following comments for this study. 

Comments: 

  • Although authors have tried to establish the role of adipose-derived macrophages (especially metabolically activated macrophages) in the development of adipocytes’  iron overload due to its iron handling machinery and related proteins. However, I was curious to know authors tried to establish the role of ATMs but no other immune cells and non-immune cells in adipocyte iron overload and disease progression?

Response: The reviewer is correct that there are many immune cells in adipose tissue that are important for homeostasis and that contribute to adipose tissue inflammation in obesity. We chose to focus on macrophages for several reasons. 1) They are the immune cells most important for systemic iron homeostasis. 2) Even at the tissue level, in other settings, macrophages are the cell type shown to have a role in tissue homeostasis – for example in wound healing. 3) Our previous work has highlighted a unique population of adipose tissue macrophages that are suggested to play a role in adipocyte iron homeostasis. We revised the Introduction to highlight this rationale.

2)    In figure 1 representation is not good for qPCR data. The font size is very small which makes reader too difficult to read the labeling. I would suggest increasing the Font size.

Response: Thank you for this comment – we have increased the size of the font as much as we can.

3)    In Figure, WB data and qPCR data is not matching. The Authors have shown that Iron regulating genes including ftl1, and cd163 were increased in M2 macrophages in qPCR data, however, expression of these genes increased in the M1 microphase shown in WB. This is contracting data. Along with this, there is no significant change in the heme oxygenase gene in M1 macrophages, however, there is a significantly increased expression of HO protein in M1 macrophages. In addition to this, there is no quantification for WB.

Response: We acknowledge that the qPCR data and Western blot data do not match for every protein. This is not uncommon, as iron protein concentrations are also strongly controlled at the post-transcriptional level by the iron response protein system. We have now included a brief description of why this discrepancy may exist in the Discussion section of the manuscript. The quantification of the Western blots shown in Figure 1C is in Figure 1D.

4)    Authors are talking about Figures in the text but there is no figure 6. This is really a big mistake. It seems that proofreading was not done by the authors. Authors should work on this.

Response: This was a terrible mistake that occurred during the upload of the manuscript. This is a very important figure and we can confirm it is included in this revised manuscript.

5)    Font size of the text is different from lines 169 -181 and 300-316. The authors should correct this.

Response: It seems that text that should have been in a regular font was italicized instead when this file was converted during the submission process. We have corrected this in the revised manuscript.

6)   Heading of material methods and Result section is not bold. It is not distinct and mixes with regular text. (For example Cell culture and cell treatments: Primary bone marrow-derived macrophage and other sections).

 Response: Thank you for pointing this out. We had been following the template provided, but we agree. All subheadings are now in bold and italicized font to make sure they stand out.

7)    Typo error in text line 147 (plasma Vfree Hgb compared…)

Response: Thank you for noting this. We have corrected this mistake.

Reviewer 2 Report

This paper titled as “An iron refractory phenotype in obese adipose tissue macrophages leads to adipocyte iron overload” is interesting. This manuscript could be considered for publication in IJMS after major revising.

My comments are as follow:

  1. The author needs to check the whole text including grammar, punctuation, writing, et al. There are too many errors.
  2. The title of manuscript does not need punctuation marks.
  3. The objectives of this study should be summarized in the abstract.
  4. Please compare your data with previous studies in the discussion section.

Author Response

This paper titled as “An iron refractory phenotype in obese adipose tissue macrophages leads to adipocyte iron overload” is interesting. This manuscript could be considered for publication in IJMS after major revising.

My comments are as follow:

  1. The author needs to check the whole text including grammar, punctuation, writing, et al. There are too many errors.

Response: Thank you for noting this. We have read through it very carefully and corrected all errors we could find.

  1. The title of manuscript does not need punctuation marks.

Response: We have removed the period from the title.

  1. The objectives of this study should be summarized in the abstract.

Response: We have revised the abstract accordingly.

  1. Please compare your data with previous studies in the discussion section.

Response: We have updated the Discussion accordingly.

Reviewer 3 Report

Minor points:

- the text refers to figure 6 A-D but this figure is not provided in the uploaded text;

- the Authors refer to Supplementary figures and table 1 but this material is not accessible to the reviewer;

- lines 171 to 181 should be normal text (not italic);

- lines 300 to 316 should be normal text (not italic).

Author Response

Minor points:

  • The text refers to figure 6 A-D but this figure is not provided in the uploaded text.

Response: This was a terrible mistake that occurred during the upload of the manuscript. This is a very important figure and we can confirm it is included in this revised manuscript.

  • The Authors refer to Supplementary figures and table 1 but this material is not accessible to the reviewer.

Response: I am not sure why that is the case – we did submit the Supplementary Figures and Table 1.

3) Lines 171 to 181 should be normal text (not italic); Lines 300 to 316 should be normal text (not italic).

Response: Thank you, we have corrected this.

Round 2

Reviewer 1 Report

In the revised manuscript, the authors have made significant changes and incorporated reviewer suggestions. However, further studies can be proposed to determine the underlying mechanism for iron loading in adipocytes and disease progression. What is the mode of transport of iron from macrophage to adipocytes? 

Author Response

We had this paragraph in the Discussion already. Will that suffice to answer your concern?

A more interesting interpretation of the observation that the differentiated cells in co-culture do not take up iron from the media is that there is an active selection bias in the cells in the co-culture to acquire iron from each other rather than from other iron rich sources such as the media. For example, iron can be transferred in exosomes [40], acidic lysosomes [41], or extruded mitochondria [42], precluding the need for dedicated protein chaperones. Although there is no evidence that adipocytes produce siderophores, this is another intriguing potential mechanism for iron transfer between these two cell types. Future studies are needed to distinguish between these possibilities. 

Reviewer 2 Report

The authors addressed all my comments.

Author Response

thank you.